# Hardness of Approximation for Langton's Ant on a Twisted Torus

**Takeo Hagiwara [1,†,‡] and Tatsuie Tsukiji [2,*,‡]** 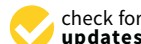

[1]  Department of Science and Engineering, Tokyo Denki University, Tokyo 120-8551, Japan; 21404@ms.dendai.ac.jp
[2]  Graduate School of Advanced Science and Technology, Tokyo Denki University, Tokyo 120-8551, Japan
[*]  Correspondence: tsukiji@mail.dendai.ac.jp
[†]  Current address: Oaza Ishizuka, Hikigun Hatoyama-tyou, Saitama 350-0394, Japan.
[‡]  These authors contributed equally to this work.

**Abstract:** Langton's ant is a deterministic cellular automaton studied in many fields, artificial life, computational complexity, cryptography, emergent dynamics, Lorents lattice gas, and so forth, motivated by the hardness of predicting the ant's macroscopic behavior from an initial microscopic configuration. Gajardo, Moreira, and Goles (2002) proved that Langton's ant is PTIME -hard for reachability. On a twisted torus, we demonstrate that it is PSPACE hard to determine whether the ant will ever visit almost all vertices or nearly none of them.

**Keywords:** approximation hardness; highway conjecture; Langton's ant; PSPCE hard; twist torus

## 1. Introduction

In 1986, Chris Langton proposed an artificial life on square lattice $\mathbb{Z} \times \mathbb{Z}$ [1–5] where each vertex is either to-right (white) or to-left (black). An *ant*, represented by an arrow in Figure 1, travels along edges with unit speed and four possible directions. It turns to either right or left, corresponding to the vertex's white or black color it is heading and switches the color simultaneously. Since then, it is one of the most widely studied cellular automata in many fields, for example, discrete propagation kinetics [6–8], Lorentz lattice gas [9–11], particle diffusion dynamics [12,13], computational complexity [14–16], emergent dynamics [17,18], and cryptography [19–23].

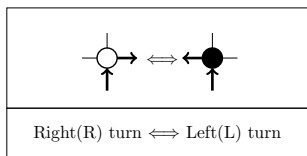

Right(R) turn ⟺ Left(L) turn

**Figure 1.** The rule of the ant's move.

This paper's motivation is the following chaotic trajectory of Langton's ant. Placed on the all-white vertices, the ant stays within $45 \times 48$ area for about 10,000 moves unpredictably, and then it suddenly starts dancing a *highway* of 104 steps drawn in Figure 2 and moving diagonally in speed $\sqrt{2}/52$ [13]. After decades of exploration, this phenomenon has been taking place with no exception, giving rise to a famous conjecture—the highway must eventually appear regardless of any initial configuration with finite support. If so, the ant's long-run behavior is predictable except for reflection and transposition. Grosfils, Boon, Cohen, and Bunimovich [12] developed diffusion dynamics proving the highway conjecture and measuring the diffusion speed on triangular lattice. Meanwhile, Gajardo, Moreira, and Goles (GMG, [14]) constructed a finite configuration to evaluate a given boolean

circuit. We note that these theoretical results fundamentally affected the related fields. For example, suppose that Langton's ant is a particle system where a finite initial configuration and its consequent macroscopic quantities are the only observables. If highway conjecture holds, macroscopic observables are fixed constants regardless of the initial state. In any case, they must have some calculation from the initial finite configuration since the system is deterministic. GMG demonstrates that calculus PSPACE-hard if macroscopic quantities rely on an ant's microscopic behavior called reachability.

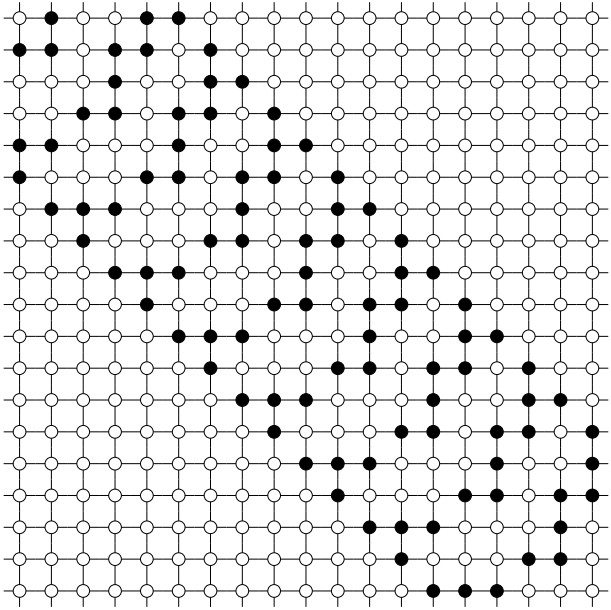

**Figure 2.** A highway.

This paper aims to strengthen GMG in both computational complexity and highway conjecture. Computational complexity measures time and space (memory) that modern computers, that is, Turing machines, must spend for solving a given computational problem (see computational complexity textbooks, for example, Reference [24]). GMG proved that the ant's trajectory could simulate any computation spending a polynomial ($n^c$ for some constant $c \geq 1$) time for any bit-length $n$ instance. GMG constructed a boolean circuit evaluator since circuit evaluation is a PTIME-complete problem, that is, simulating any polynomial-time calculation. Similarly, QCNF is a PSPACE-complete problem to execute any polynomial-space computation [25]. QCNF evaluates a given Conjunctive Normal Form (CNF) with bounded quantifiers. For example, QCNF evaluates $\forall x_1, \exists x_2, (x_1 \vee \neg x_2) \wedge (\neg x_1 \vee x_2) = \text{TRUE}$, since for any $x_1 \in \{\text{TRUE, FALSE}\}$, there exists $x_2 \in \{\text{TRUE, FALSE}\}$, making the 2-term 2-variable CNF $(x_1 \vee \neg x_2) \wedge (\neg x_1 \vee x_2) = \text{TRUE}$. PSPACE-complete problems take exponential time for any known algorithms, although P $\neq$ PSPACE is merely a conjecture. We note that GMG constructed a universal Turing machine but using an infinite number of black vertices for the initial configuration.

This paper aims to obtain the first result that the ant's trajectory is unpredictable, even for a macroscopic measurement [3]. Let $\mathbb{Z}_n = \{-\lfloor \frac{n-1}{2} \rfloor, \cdots, -1, 0, 1, \cdots, \lceil \frac{n-1}{2} \rceil\}$. For a coloring $\theta_n$ with black vertices in only $\mathbb{Z}_n \times \mathbb{Z}_n$, let $f_{\theta_n}(N, T)$ be the number of vertices in $\mathbb{Z}_N \times \mathbb{Z}_N$ that the ant starting from the origin will visit after $T$ steps. We call $\mathbb{Z}_n \times \mathbb{Z}_n$ *microscopic* field, while $\mathbb{Z}_N \times \mathbb{Z}_N$ *macroscopic* one. If highway conjecture holds, we have $\forall \theta = \{\theta_n\}_n, \rho_1(\theta) = 11/\sqrt{2}$ for $\rho_c(\theta) := \lim_{n \to \infty} \lim_{N \to \infty} \lim_{T \to \infty} f_{\theta_n}(N, T)/N^c$. However, we do not even know $\forall \theta, \rho_2(\theta) < 1$, that is, the ant may visit almost all vertices for some $\theta$. The only known previous result is $\forall \theta, \rho_1(\theta) > 0$, that is, the ant's trajectory must always be unbounded [10]. Although $\rho_2(\theta)$ measures $N^2$-order convergence rather than more accurate $N$-order one required in highway conjecture, its impact is substantial on the related fields. For example, if $\forall \theta, \rho_2(\theta) = 0$, mathematical analysis may have a chance to prove highway conjecture on square lattice, as they have done in triangular lattice [9,12]

and one-dimensional lattice [7,12]. If $\exists\theta, \rho_2(\theta) = 1$, cryptographic applications [19–23] may find finite configurations producing infinitely long pseudo-random bits generated by the ant's trajectory.

This paper will show that $\rho_2(\theta)$ is hard to approximate on a *twisted torus*. A torus is the standard boundary condition to extend a coloring $\phi$ on $\mathbb{Z}_N \times \mathbb{Z}_N$ to $\mathbb{Z} \times \mathbb{Z}$ by $\phi(x,y) = \phi(x+N,y) = \phi(x,y+N)$ for $(x,y) \in \mathbb{Z} \times \mathbb{Z}$. Long-run computer simulation of square lattice cellular automata usually takes this topology [19,20,22]. An *h-twist* (*h*/*N*-pitch) of the torus is $\phi(x,y) = \phi(x+N,y) = \phi(x+h,y+N)$. It appears in related models, for example, square-lattice Ising models (Langton's ant is a typical one) [26,27]. When $h = \pm 11$, $\rho_2(\theta) = 1$ may occur even under highway emergence, as described in Figure 3. For this specific case, we can prove the following theorem.

**Theorem 1.** *On $\pm 11$-twist torus, it is PSPACE hard to distinguish between $\rho_2(\theta) = 1$ and 0.*

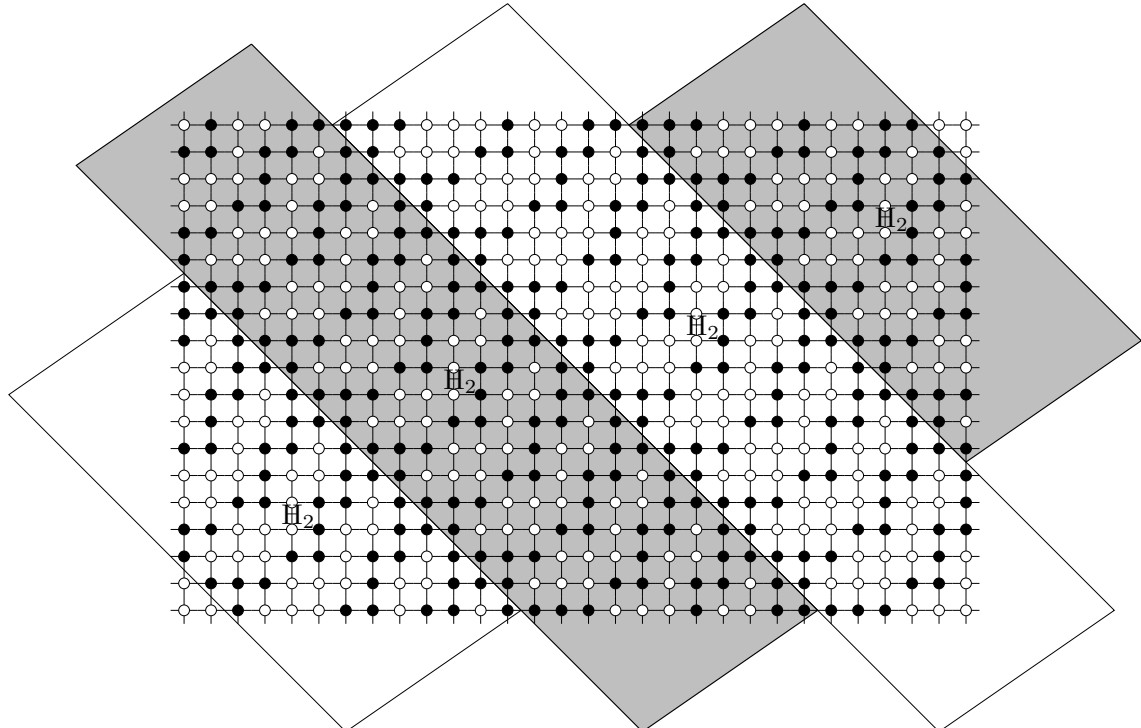

**Figure 3.** A screwing highway.

It states that modern computers cannot predict whether the ant on the twisted torus will ever visit almost all vertices or nearly none of them, even under the promise that either of these cases must occur. Notice that $\rho_2(\theta)$ is a time-independent measurement, counting the ever visiting vertices. Our proof will measure the *period* that an initial configuration of the cellular automaton appears again, too. Any closed loop of Langton's ant's propagating configurations must contain the initial one [6]. Let $g_{\theta_n}(N)$ be the period of Langton's ant on $\mathbb{Z}_N \times \mathbb{Z}_N$ with an initial coloring $\theta_n$ on $\mathbb{Z}_n \times \mathbb{Z}_n$. Let $\xi_c(\theta) := \lim_{n\to\infty} \lim_{N\to\infty} g_{\theta_n}(N)/N^c$.

**Theorem 2.** *On $\pm 11$-twist torus, it is PSPACE hard to distinguish between $(\rho_2(\theta), \xi_2(\theta)) = (1, \frac{208}{11})$ and $(0,0)$.*

Consequently, no computer-aided analysis can calculate Langton's ant's macroscopic quantities from an initial finite configuration in polynomial time, unless P = PSPACE. We note that even under the (twisted) tours's boundary condition, Lorentz lattice gas can measure the ant's diffusion speed for triangular lattice [9], particle diffusion dynamics can do it for both triangular and one-dimensional lattices [12], and discrete propagation kinetics the diffusion speed or period of the ant's rules for one-dimensional lattice [7]. Theorem 2 indicates that these currently known methods cannot measure them for square lattice.

*A Proof Outline*

Figure 4 reduces QCNF to Langton's ant problem of distinguishment between $(\rho_2(\theta), \xi_2(\theta)) = (0,0)$ and $(1, \frac{208}{11})$. It maps a given QCNF instance $\Phi = \{\Phi_n\}$, say $\Phi_{n_0} = \forall x_1, \exists x_2, (x_1 \vee \neg x_2) \wedge (\neg x_1 \vee x_2)$ for a sufficiently large $n_0$ to encode it on $\mathbb{Z}_{n_0} \times \mathbb{Z}_{n_0}$, to a coloring $\theta(\Phi) = \{\theta_n(\Phi_n)\}$ on microscopic field. The ant starting from the *port* (i.e., edge) $S$ of TURN-A gate walks along a line and gets into the $G[\Phi]$ *chip* (a collection of gates) at the entrance port $I$. The ant evaluates $\Phi$ and gets out from either the exit port F if it is FALSE or T if it is TRUE. In the former case, the ant reaches TURN-B gate. Since each TURN gate *reverses* the ant (i.e., putting the ant on the same port in the opposite direction), it must repeat a closed-loop between TURN-A and TURN-B gates due to *time-reversibility*; the reversed ant must travel back and rewind all steps it has taken in the time-reversing order. This loop holds the ant on microscopic field and thus makes a reduction $\Phi = \text{FALSE} \Rightarrow (\rho_2(\theta(\Phi)), \xi_2(\theta(\Phi))) = (0,0)$. Similarly, the latter case induces another loop between TURN-A and TURN-C gates, passing through Hamiltonian-tour gate on the way, which drives the ant to visit almost all vertices in macroscopic field. It yields $\Phi = \text{TRUE} \Rightarrow (\rho_2(\theta(\Phi)), \xi_2(\theta(\Phi))) = (1, \frac{208}{11})$, where $\frac{208}{11} = 4/11 \cdot \sqrt{2}/(\sqrt{2}/52)$. Since these TURN gates take binary states flipping at the ant's every visit, the ant travels Figure 3's Hamiltonian-tour four times until revisiting the initial configuration. Each Hamiltonian-tour consists of $(N - O(n))/11$ rounds of length-$\sqrt{2}N$ diagonals, taking $(N - O(n))/11 \cdot \sqrt{2}N/(\sqrt{2}/52)$ time since the ant's speed is $\sqrt{2}/52$ [13].

Our previous paper [15] has already provided a coloring for the $G[\Phi]$ chip to evaluate a QCNF formula $\Phi$, but using *gray vertices* at which the ant goes straight without changing the color [10]. Since it has permitted only a single gray vertex at each TURN gate and nowhere else, we are enough to exhibit only a new TURN gate with no gray vertex. However, open square lattice topology prohibits it because it tessellates the vertices of $\mathbb{Z} \times \mathbb{Z}$ into chess-board symmetry of $\{H, V\}$-polarity: Every H-vertex (resp. V-vertex) permits the ant to come Horizontally (resp. Vertically) and go vertically (resp. horizontally). Even-twist torus preserves $\{H, V\}$-polarity, but odd-twist torus breaks it. If the ant living in odd-twist torus travels around vertically and comes back to the initial place, it can visit the world where every vertex possesses the opposite polarity. Figure 5 realizes a TURN gate in virtue of this polarity switching phenomenon. The ant enters at the port $S$, passing through FL (FLipper) gate , and reaching HW (HighWay) gate. The ant gets on Figure 2's highway and travels towards a $\sqrt{2}N$ away destination place in a diagonal direction. The odd-twist torus's topology brings the ant to a new world possessing the same address but the opposite polarity. Accordingly, after getting off the highway, the ant passes through the FL gate again and gets out the TURN gate from the same port $S$ that it has entered but in the opposite direction.

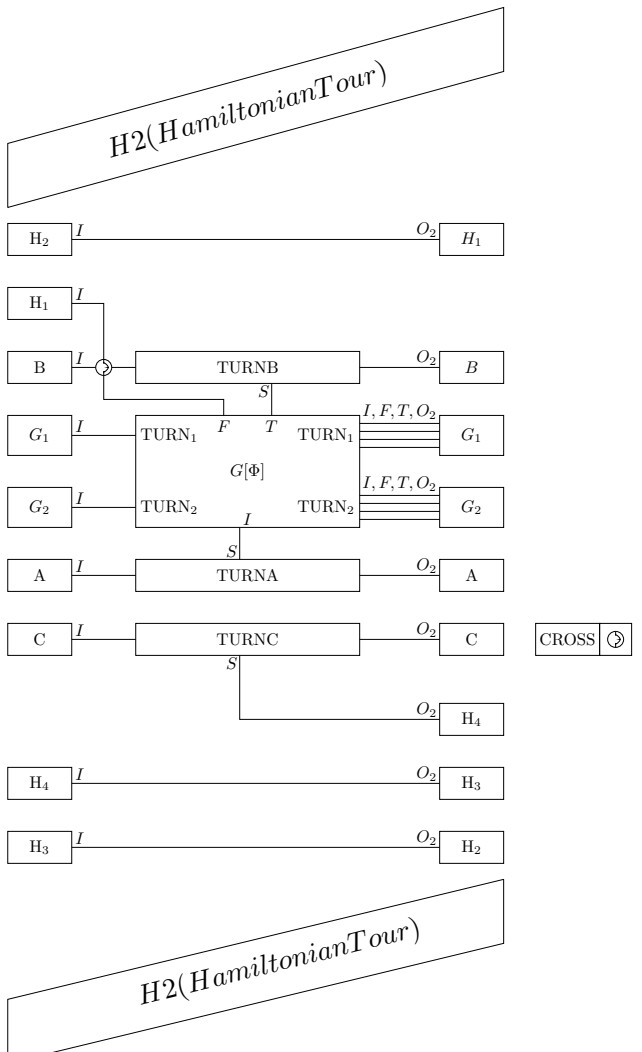

**Figure 4.** An ant's travel for QCNF .

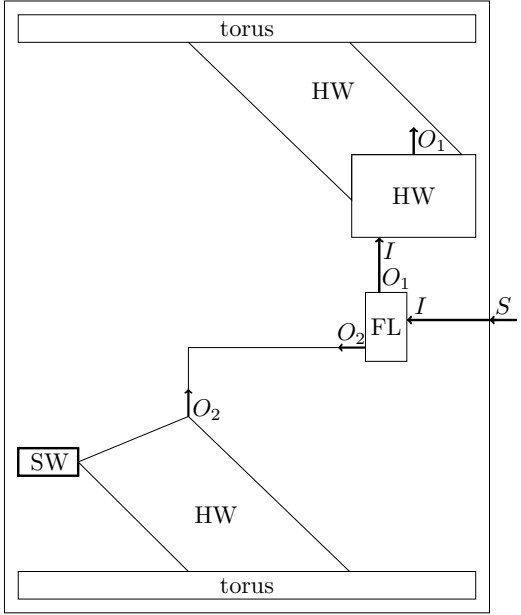

**Figure 5.** TURN .

A Hamiltonian-tour gate to visit almost all vertices is easy to realize: Let the ant get on a highway at the entrance port I of top-left $H_2$ in Figure 4. It brings the ant to a Hamiltonian tour along screwing diagonals of Figure 3: Each diagonal transports the ant by a horizontal and vertical distance $N$, bringing it back to the same address but a horizontal displacement $\pm 11$ due to the twisted torus topology. This gap number 11 fits the highway width, providing an exhaustive, mutually exclusive vertex cover. The highway must assume white color of the underlying vertices, so it remains at most $n^2$ vertices inside microscopic field unvisited. Figure 6 paves gates along the border between microscopic and macroscopic fields to generate consecutive multiple highways $H_1, H_2, \ldots, H_4$ surrounding the $G[\Phi]$ chip. The ant must pass through all these highways in order: It gets on each $H_i$ at the entrance port $I$ of Figure 6 placed in Figure 4's left $H_i$, making a trip to the right $H_i$, getting off there at the exit port $O_2$, and walking along a line guiding to the next left $H_{i+1}$'s entrance port.

We follow an analysis of Reference [16] in making efficient embedding of Figure 4's cellular automaton to the square lattice, although it allowed using gray color freely in CROSS gates. Reference [15] prohibits it there, but their CROSS gate was large and complex. This paper will maintain the efficiency of Reference [16] by providing a new construction of CROSS gate having size $9 \times 9$ in Figure 7.

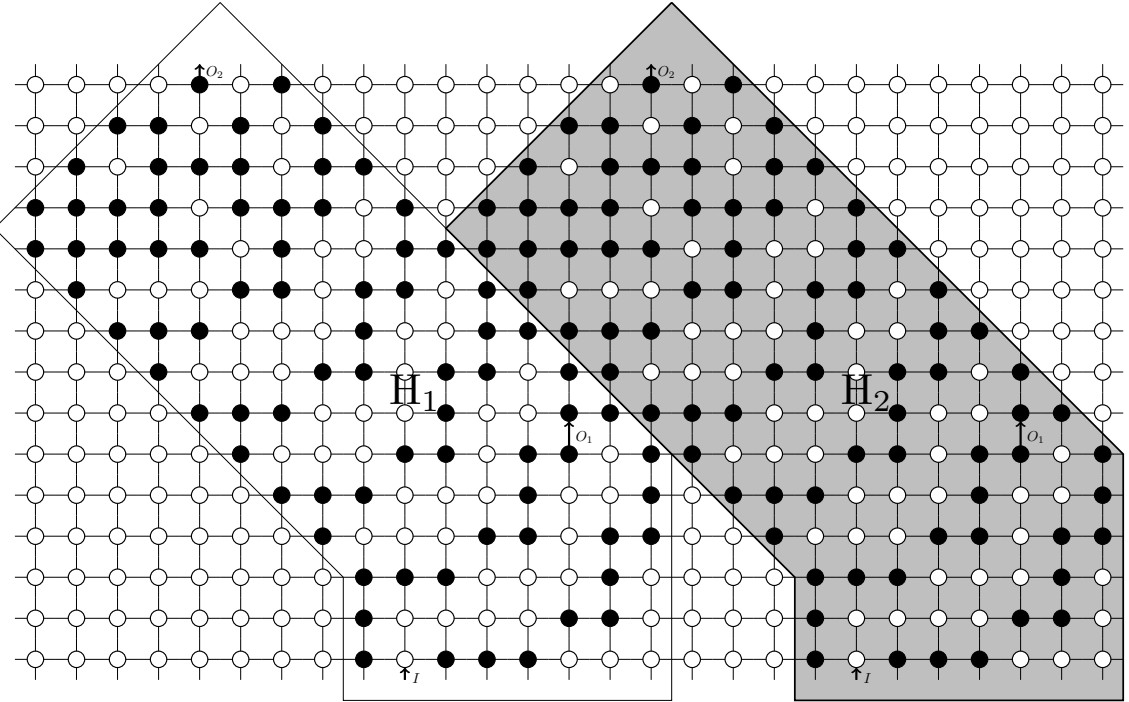

**Figure 6.** Paving consecutive HW.

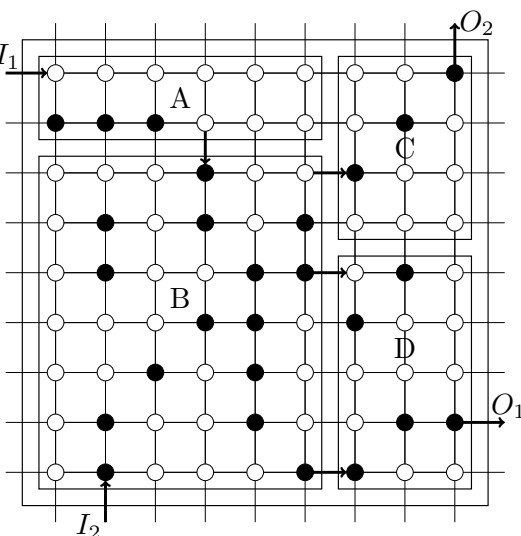

**Figure 7.** Cross gate.

## 2. Gates

A *gate* is a coloring on a limited lattice area for embedding the corresponding part of Figure 4's cellular automaton to square lattice. This section describes the necessary gates composing all parts of Figure 4. These gates must endure repeated usage under contamination by the ant's passage. Time reversibility of Langton's ant recovers the initial state: The ant passes a gate by $I \rightarrow O$, that is, entering at $I$, staying inside the gate for a while, and finally exiting at $O$, changing the gate's state $C \rightarrow D$. After that, the ant placed on $O^{-1}$, the same port $O$ but the opposite direction, must proceed as $O^{-1} \rightarrow I^{-1}$ and changes the gate's state as $D \rightarrow C$. We say that the ant's inverse trip must charge the dry gate for the next passage.

Figures 8–12 display gates and their state diagrams called Straight PATH, Bending PATH, JOIN (called CONJ in Reference [15]), S&P (Switch and Pass), and SWitch (SW in short), respectively. These state diagrams are Mealy machines outputting ant's tracks when state transitions occur. When $C \rightarrow D$ occurs, the ant must travel from $I$ to $O$ inside the gate along the *track* (a series of the ant's moves to either the Left (L) or Right (R) directions). We write it as $I[X] \xrightarrow{C/D} O[X]$ or $C[X] \xrightarrow{I/O} D[X]$. For example, PATH can stretch upwards by alternating Figure 8 with its H&D (H-symmetry and Dual, that is, flipping Horizontally and swapping C ↔ D), $I[\text{PATH}] \xrightarrow{I/O} O[\text{PATH}] \rightarrow I[\text{H&D PATH}] \xrightarrow{I/O} O[\text{H&D PATH}] \rightarrow I[\text{PATH}] \xrightarrow{I/O} O[\text{PATH}] \rightarrow \cdots$, forming a width-2 line, possibly bending to right by Figure 9. It can stretch and turn by 90 degrees to arbitrary directions by taking the symmetric and dual figures. PATH, S&P, and JOIN have the same configurations as Reference [15]. Differences from Reference [15] are parsing a S&T (Switch and Turn) gate in Reference [15] into three gates (SW, HW, FL) in Figure 5, providing a new TURN gate with no gray-color vertex, and a new CROSS gate having a much smaller size.

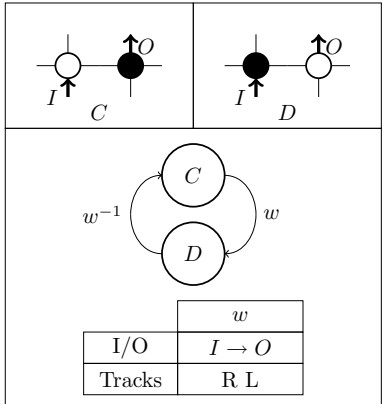

**Figure 8.** Straight PATH.

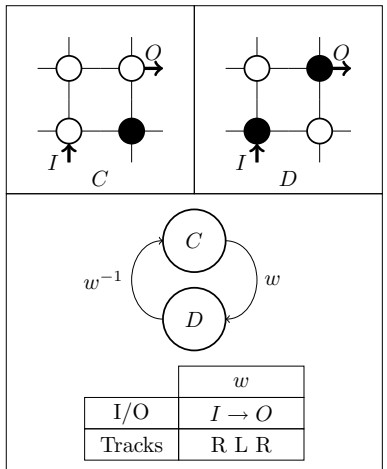

**Figure 9.** Bending PATH.

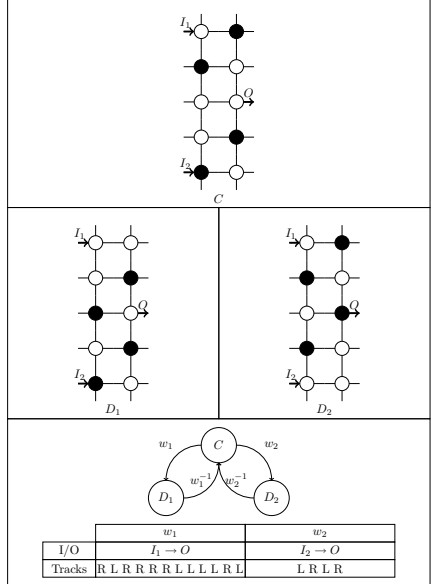

**Figure 10.** JOIN.

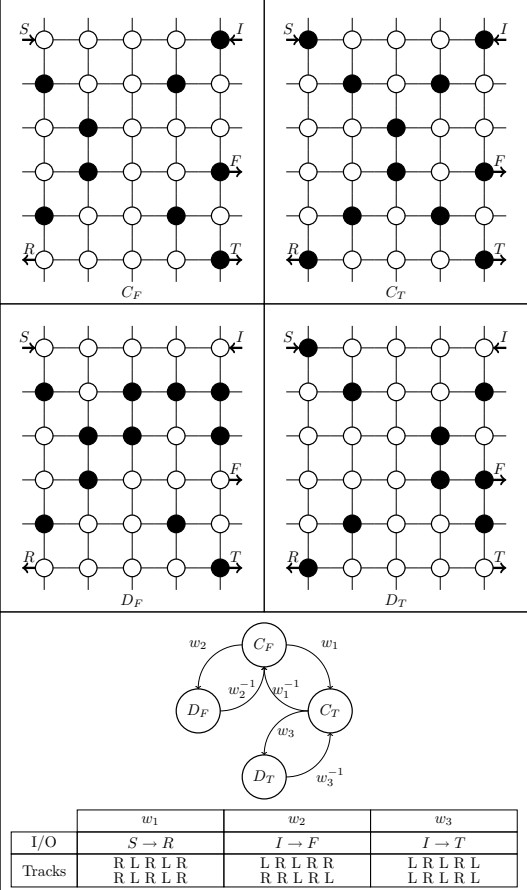

**Figure 11.** S&P.

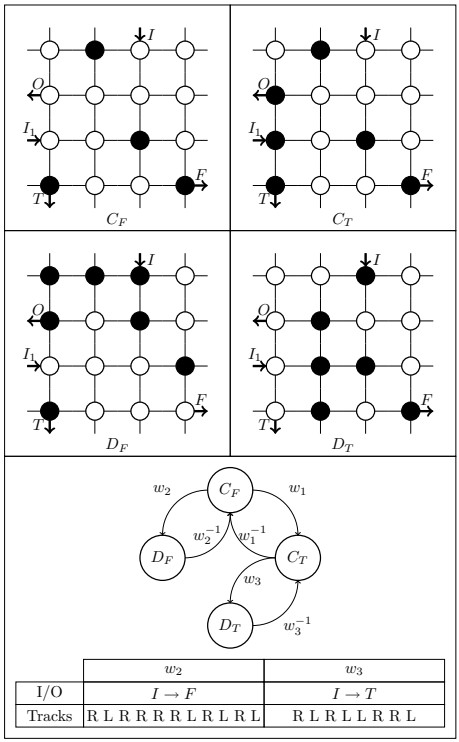

**Figure 12.** SW.

## *2.1. Turns*

Figures 12–14 present configurations of SW, FL, and HW, respectively. Any automatic method can check the correctness of these configurations. In Figure 14, the ant gets in HW at the entrance port $I$, passing through $I \to O_1$ for highway generation, riding on a highway propagation $O_1 \to O_2$, and getting out of it at the exit port $O_2$ and walking away along a path starting from $O_2$. Figure 15 paves consecutive highways where each HW attaches an SW (horizontally flipped Figure 12) to share HW's right-most two vertices with SW's left-most ones. In this manner, Figure 4 paves the $k$ TURNs ($G_1$ and $G_2$ with $k = 2$ there) for the $k$ variables of $\Phi$.

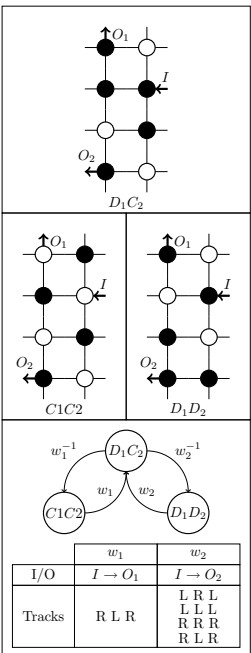

**Figure 13.** FL.

**Lemma 1.** *Every visit of the ant to Figure 5 induces* $S[TURN] \to S^{-1}[TURN]$*, accompanied by* $C_F[SW] \xrightarrow{I_1/O} C_T[SW]$ *(resp.* $C_T[SW] \xrightarrow{O^{-1}/I_1^{-1}} C_F[SW]$*) at each odd (resp. even) time visit.*

**Proof.** In each odd time visit to Figure 5, the ant gets in from the entrance port $S$, travels along

$$S \to I[\text{FL}] \xrightarrow{C_1 C_2 / D_1 C_2} O_1[\text{FL}] \to I[\text{HW}] \xrightarrow{C_1 C_2 / D_1 C_2} O_1[\text{HW}] \xrightarrow[\to I_1[\text{SW}] \xrightarrow{C_F/C_T} O[\text{SW}] \to]{D_1 C_2 / D_1 D_2} O_2[\text{HW}] \to^*$$

$O_2^{-1}[\text{FL}] \xrightarrow{D_1 C_2 / D_1 D_2} I^{-1}[\text{FL}] \to S^{-1}$, and exits from the $S$ it has entered but in the opposite direction. Here, the ant at $\to^*$ can get into the FL gate from the exit port $O_2$ since the ant lives in the opposite polarity world before and after taking the HW trip. Each even time visit induces the reversed travel

$$S \to I[\text{FL}] \xrightarrow{D_1 D_2 / D_1 C_2} O_2[\text{FL}] \to O_2^{-1}[\text{HW}] \xrightarrow[\to O^{-1}[\text{SW}] \xrightarrow{C_T/C_F} I_1^{-1}[\text{SW}] \to]{D_1 D_2 / D_1 C_2} O_1^{-1}[\text{HW}] \xrightarrow{D_1 C_2 / C_1 C_2} I^{-1}[\text{HW}] \to$$

$O_1^{-1}[\text{FL}] \xrightarrow{D_1 C_2 / C_1 C_2} I^{-1}[\text{FL}] \to S^{-1}$. Observe that the SW's state has transited as claimed.　□

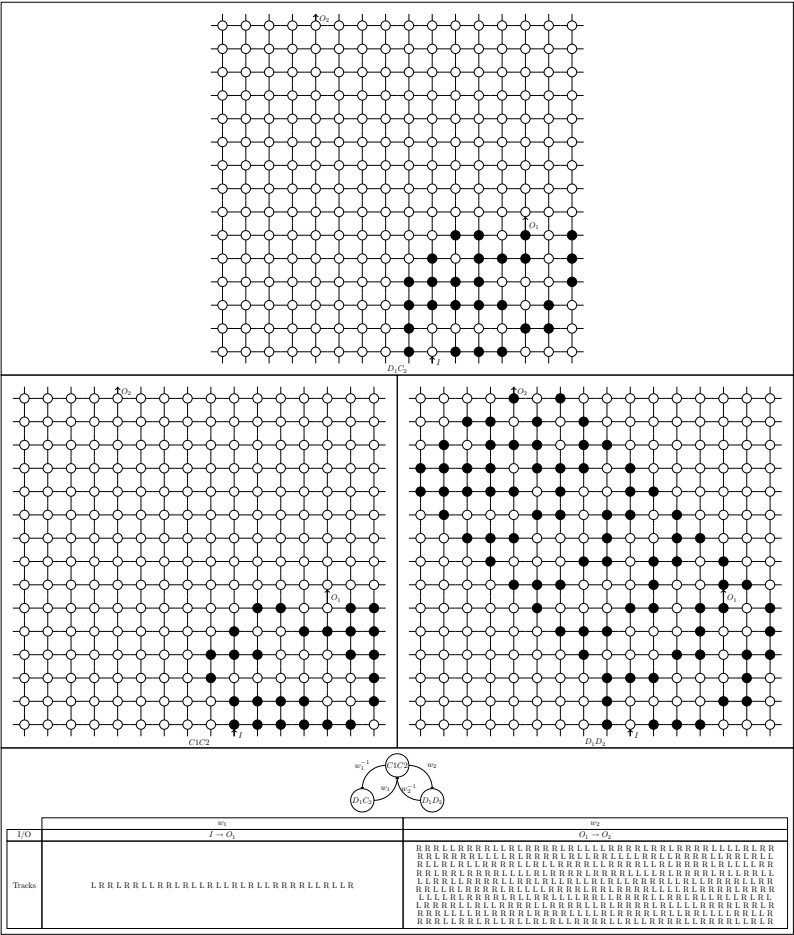

**Figure 14.** HW.

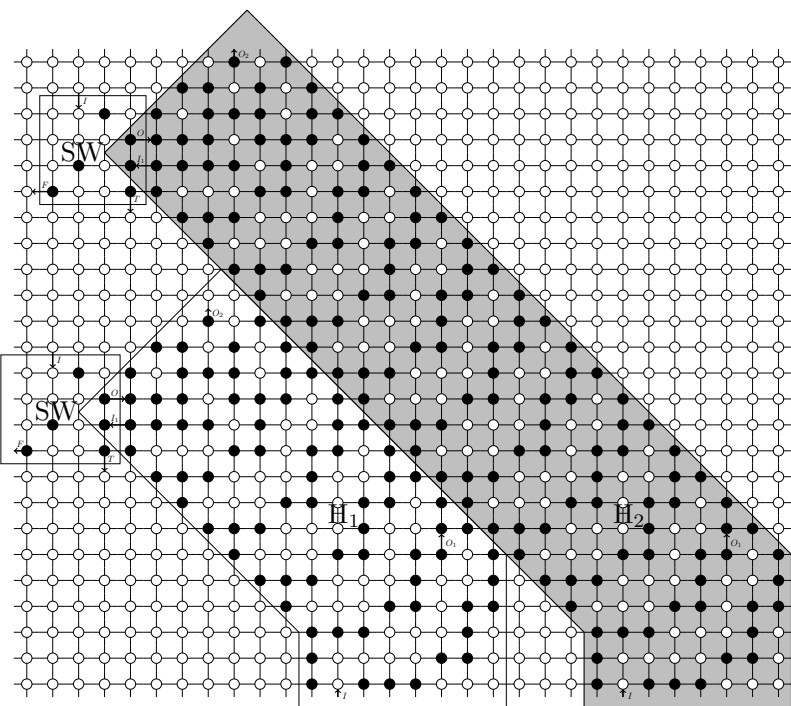

**Figure 15.** Paving consecutive HW with SW .

## 2.2. Crosses

Figure 16 draws a state diagram of Figure 7's CROSS gate constituting from 4 sub gates A–D; Figures 17–20 show their configurations. It is automatic to check their correctness.

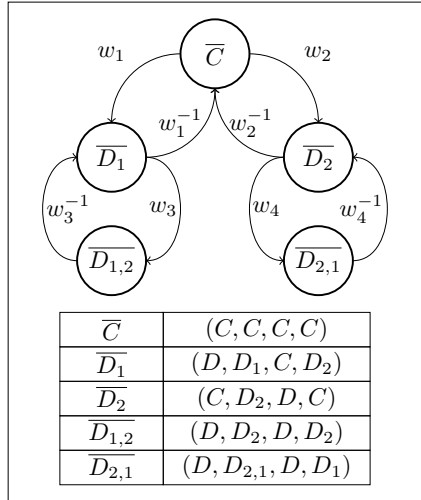

| $\overline{C}$ | $(C, C, C, C)$ |
|---|---|
| $\overline{D_1}$ | $(D, D_1, C, D_2)$ |
| $\overline{D_2}$ | $(C, D_2, D, C)$ |
| $\overline{D_{1,2}}$ | $(D, D_2, D, D_2)$ |
| $\overline{D_{2,1}}$ | $(D, D_{2,1}, D, D_1)$ |

**Figure 16.** Cross diagram.

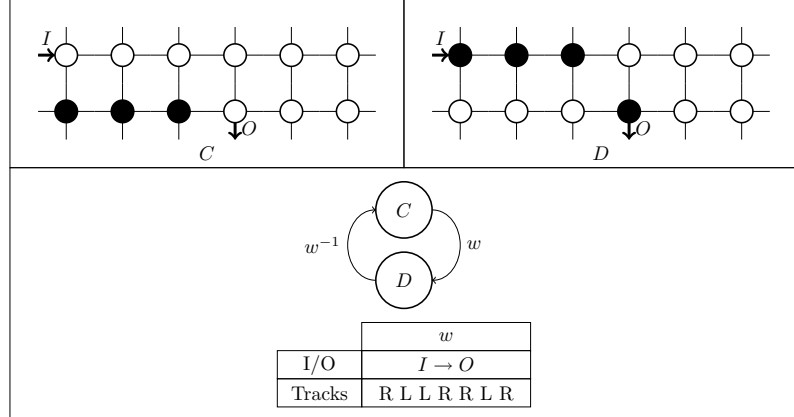

| | $w$ |
|---|---|
| I/O | $I \to O$ |
| Tracks | R L L R R L R |

**Figure 17.** Cross A.

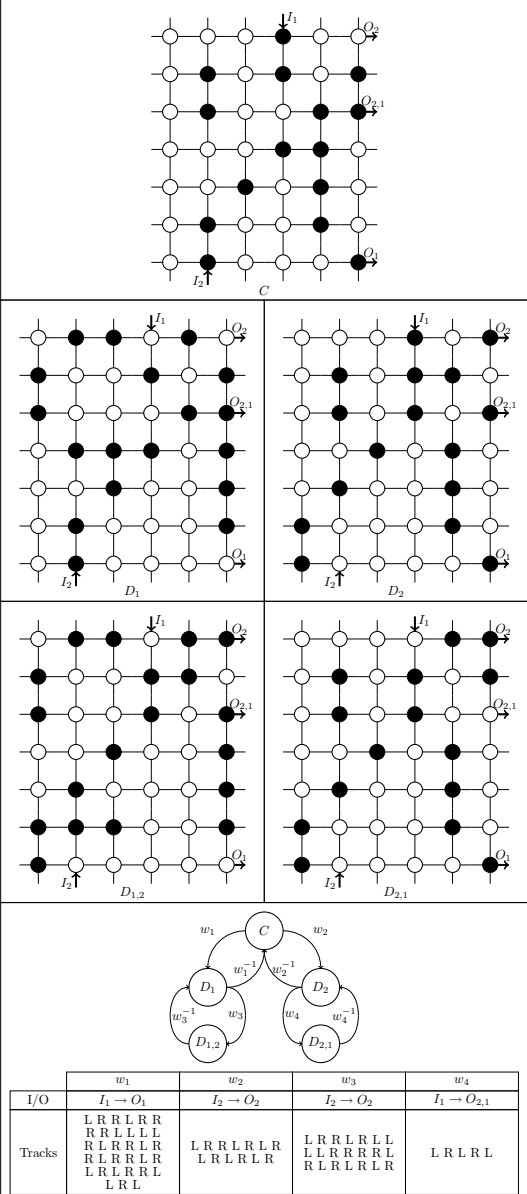

**Figure 18.** Cross B.

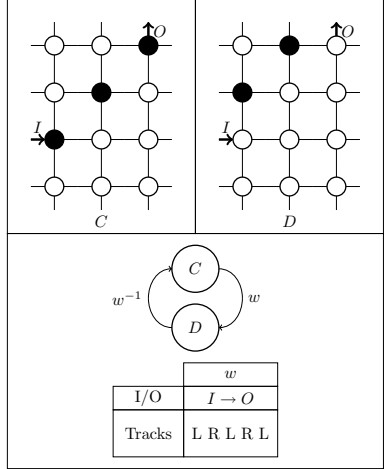

**Figure 19.** Cross C.

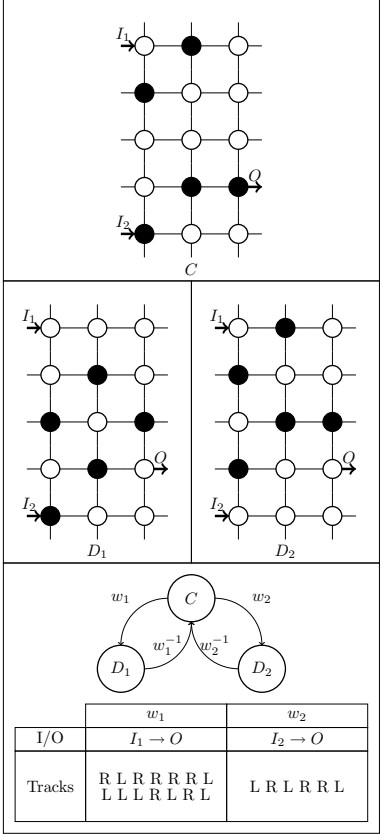

**Figure 20.** Cross D.

## 3. Lattice Embedding

A previous paper [16] transformed the $G[\Phi]$ chip to a planar cellular circuit over {S&P, S&T, JOIN, CROSS} gates by Figures 21–23 for $\Phi = \forall x_1, \exists x_2, (x_1 \vee \neg x_2) \wedge (\neg x_1 \vee x_2)$.

**Lemma 2** ([16]). *A quantified CNF formula of k boolean variables, having m clauses, and $\ell$ occurrences of the literals can be computed by a planar cellular circuit having $k + \ell$ S&P, k S&T, $\ell - 1 + k$ JOIN, and $2(\ell + m/2)^2 + 9k$ CROSS gates.*

The planar cellular circuit takes a classical Visual Representation (VR) [28,29] for square lattice embedding in Reference [16]. Notice that VR embedding have several aesthetic criteria, say draw edge as straight as possible, vertices should be evenly distributed, minimize bends in the lines. It represents each {S&P, JOIN, CROSS, SW, HW, FL} gate by a horizontal line segment and each path is connecting between them by a vertical line segment. Figure 7's CROSS gate has size $9 \times 9$ with just one port on each side, around which the ant can move by paths of width 2. Consequently, its VR embedding enjoys both horizontal and vertical sizes $9 + 2 \cdot 2 = 13$.

**Lemma 3.** *Any cellular {S&P, JOIN, CROSS, SW, HW, FL}-circuit consisting of S CROSS gates and the other $o(S)$ gates have a VR embedding on $\mathbb{Z}_{(13+o(1))S} \times \mathbb{Z}_{(8+o(1))S}$.*

**Proof.** VR embedding takes a height 13 to embed each CROSS vertex and a width 8 for the four lines of width-2 emitted from it. Since there are only $o(S)$ of the other kinds of vertices, the total height and width are no larger than $13S + o(S)$ and $8S + o(S)$, respectively. $\square$

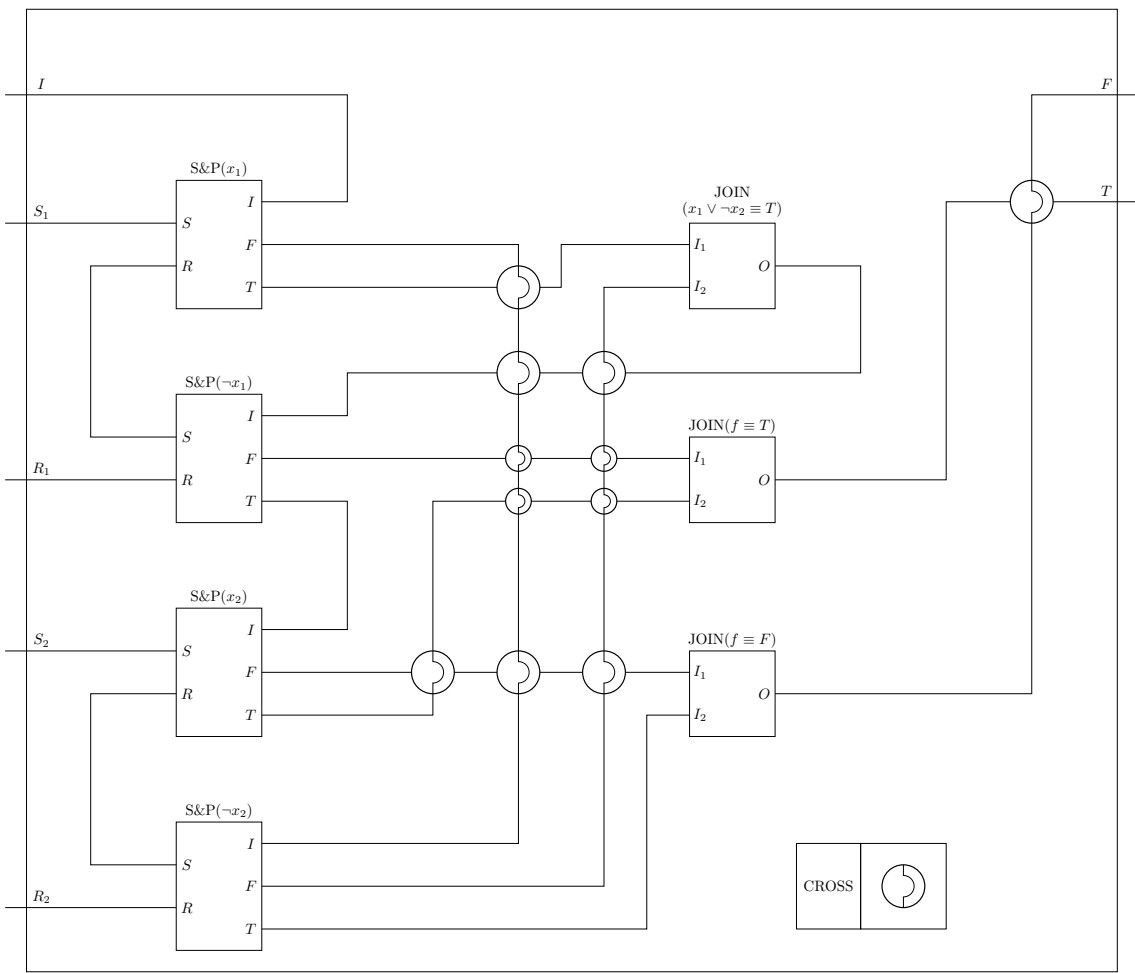

**Figure 21.** $G[(x_1 \vee \neg x_2) \wedge (\neg x_1 \vee x_2)]]$.

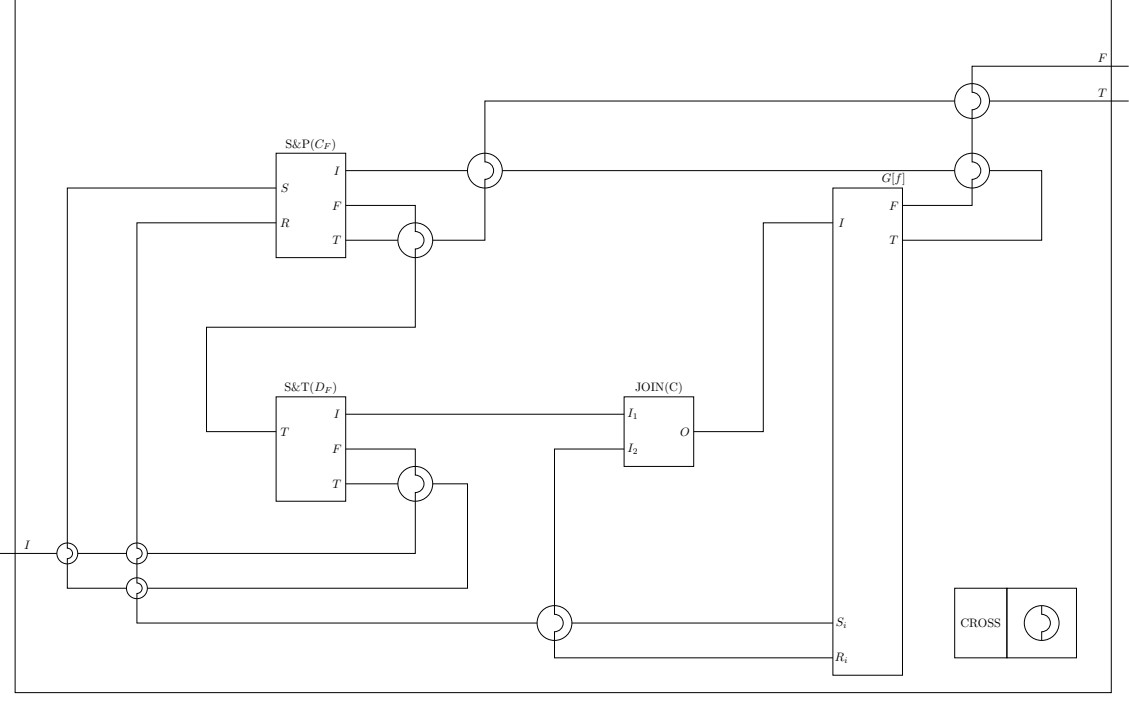

**Figure 22.** $G[(\forall x) f]]$.

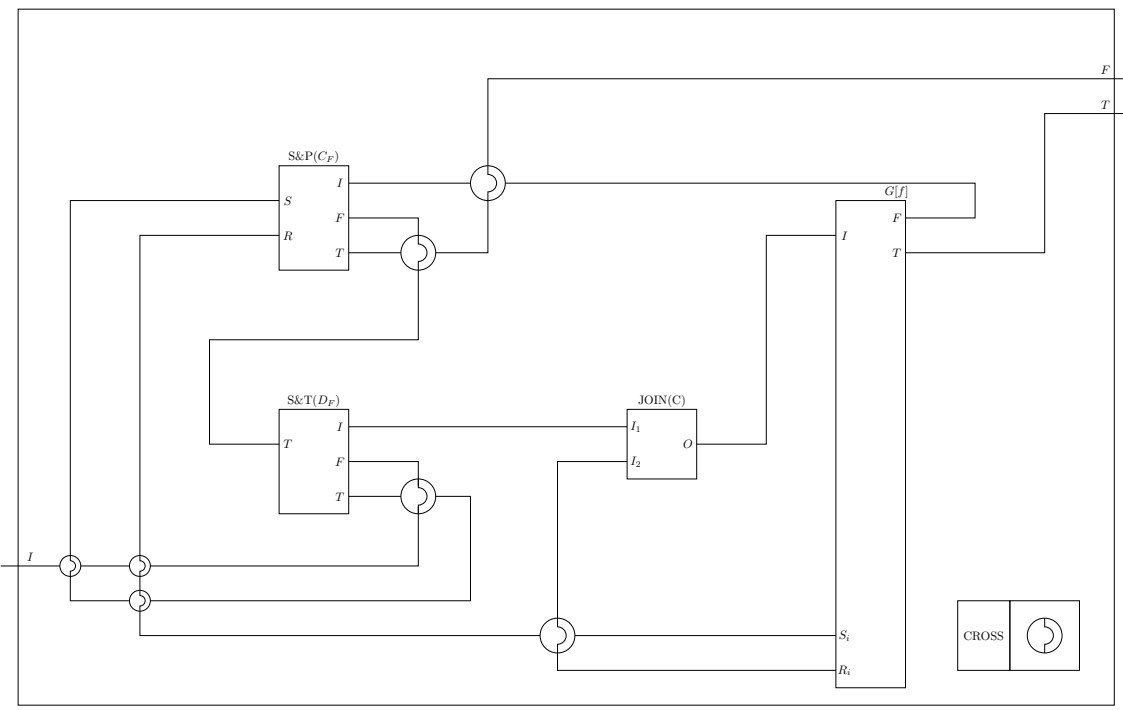

**Figure 23.** $G[(\exists x)f]$.

## 4. Hardness of Approximation for Langton's Ant

We say that a function $f : \{0,1\}^* \to \{0,1\}^*$ has a deterministic or non-deterministic computation of $t$-time and $s$-space if it has that kind of Turing machine $T$ halting in $t$ steps by using $s$ walk-tape cells such that $\forall x \in \{0,1\}^*, f(x) = T(x)$. A language $L : \{0,1\}^* \to \{0,1\}$ has a closed QCNF expression if there exists a sequence of closed QCNF expressions $\{\Phi(x_1,\ldots,x_k)\}_{k\geq 1}$ such that $\forall k, \forall x \in \{0,1\}^k, L(x) = \Phi(x_1,\ldots,x_k)$; a closed QCNF is a QCNF with all $k$ variables quantified. CNF's size sums the numbers of variables in the terms; QCNF's size is its CNF's size.

**Theorem 3** ([25]). *Let $s$ be any integer larger than the number of input bits. Any language having a non-deterministic computation using $s$ space admits a closed QCNF representation of size $cs^2$ for some universal constant $c > 0$.*

The next theorem demonstrates Theorems 1 and 2 by a reduction from the above theorem.

**Theorem 4.** *Let $h = \pm 11$ and $s^4 \ll n \ll N$. A given closed QBF $\Phi$ of size $cs^2$ is transformable by a deterministic computation of $n^{O(1)}$ time and $O(\log n)$ space to a coloring $\theta_n(\Phi)$ on $\mathbb{Z}_n \times \mathbb{Z}_n$ of h-twist torus $\mathbb{Z}_N \times \mathbb{Z}_N$, such that if $\Phi = FALSE$ then $N^2 - n^2 \leq \lim_{T\to\infty} f_{\theta_n}(N,T) \leq N^2$ and $\frac{208}{11}(N^2 - n^2) \leq g_{\theta_n(\Phi)}(N) \leq 208(\frac{1}{11}N^2 + 2^{cs^2}N)$, else $\lim_{T\to\infty} f_{\theta_n(\Phi)}(N,T) \leq \sqrt{n}N$ and $g_{\theta_n(\Phi)}(N) \leq 208 \cdot 2^{cs^2} N$.*

**Proof.** Lemmas 2 and 3 embed the $G[\Phi]$ chip to $\mathbb{Z}_{\tilde{n}} \times \mathbb{Z}_{\tilde{n}}$ for $s^4 \ll \tilde{n} \ll n$ in the claimed time and space due to VR embedding algorithms [28,29]. A previous paper [15] proved that if $\Phi$ is TRUE (resp. FALSE), then the ant getting in $G[\Phi]$'s entrance port $I$ must get out from the exit port $T$ (resp. $F$). As explained by Figure 4 in Section 1, if $\Phi$ is TRUE, the ant repeats $S^{-1}[\text{TURN-A}] \to I[G[\Phi]] \xrightarrow{C/D} S[G[\Phi]] \to S[\text{TURN-B}] \xrightarrow{C/D} S^{-1}[\text{TURN-B}] \to T^{-1}[G[\Phi]] \xrightarrow{D/C} I^{-1}[G[\Phi]] \to S[\text{TURN-A}] \xrightarrow{C/D} S^{-1}[\text{TURN-A}] \to \cdots \to S[\text{TURN-A}] \xrightarrow{D/C} S^{-1}[\text{TURN-A}]$. The ant visits only $\tilde{n}^2$ vertices of the $G[\Phi]$ chip plus $11N$ of each HW gate for the $k$ variables of $\Phi$, yielding $f_{\theta_n}(N,T) \leq \tilde{n}^2 + k \cdot 11N \ll N\sqrt{n}$. The ant switches the vaiables' values for $2^{k+2}$ times since each of the four trips between TURN-A and TURN-B may consult at most $2^k$ bit patterns to evaluate QCNF. Switching a variable takes

at most $52N$ steps for tripping HW in the TURN gate. Between them, the ant travels the same polarity, so each S&P, SW, and CROSS gate allow twice passages, and the other gates only once (see their state diagrams). Since the $G[\Phi]$ chip contains at most $O(s^4)$ gates and $\tilde{n}^2$ total path length, $g_{\theta_n}(N) \leq 2^{k+2} \cdot (52N + O(s^4) + \tilde{n}^2) \ll 208 \cdot 2^{cs^2} N$.

If $\Phi$ is FALSE, $S^{-1}[\text{TURN-A}] \rightarrow I[G[\Phi]] \xrightarrow{C/D} F[G[\Phi]] \rightarrow H_1[I] \xrightarrow{C/D} H_1[O_2] \rightarrow \cdots \rightarrow H_j[I] \xrightarrow{C/D} H_j[O_2] \rightarrow S[\text{TURN-C}] \xrightarrow{D/C} S^{-1}[\text{TURN-C}] \rightarrow H_j[O_2^{-1}] \xrightarrow{D/C} H_j[I_2^{-1}] \rightarrow \cdots \rightarrow H_1[O_2^{-1}] \xrightarrow{D/C} H_1[I_2^{-1}] \rightarrow F^{-1}[G[\Phi]] \xrightarrow{D/C} I^{-1}[G[\Phi]] \rightarrow S[\text{TURN-A}] \xrightarrow{C/D} S^{-1}[\text{TURN-A}] \rightarrow \cdots \rightarrow S[\text{TURN-A}] \xrightarrow{D/C} S^{-1}[\text{TURN-A}]$. Here, $j \leq \tilde{n}$ ($j = 4$ in Figure 4) is the number of Figure 6's HW gates surrounding the $G[\Phi]$ chip, one of which ($H_2$ in Figure 4) becomes a Hamiltonian tour. Consequently, $\lim_{T \to \infty} f_{\theta_n}(N, T) \geq N^2 - \tilde{n}^2 > N^2 - n^2$, and $g_{\theta_n}(N) > (N^2 - n^2) \cdot \frac{208}{11}$. An upper bound of $g_{\theta_n}(N)$ is the Hamiltonian tour time-bound $\frac{208}{11} N^2$ plus the $G[\Phi]$'s evaluation $208 \cdot 2^{cs^2} N$.

The only difference between Figure 4's cellular automaton and the previous one in Reference [15] is parsing each $k$ S&T gate inside Figures 22 and 23 to Figure 5's (SW, HW, FL) gates, and moving out both HW and SW to the outside but remaining the FL there. Observe that the SW gate entails three lines connecting $I[\text{SW}]$–$I_1[\text{JOIN}]$, $F[\text{SW}]$-$I_1[\text{CROSS}]$, and $T[\text{SW}]$–$I_2[\text{CROSS}]$, and the HW gate two lines $I[\text{HW}]$–$O_1[\text{FL}]$ and $O_2[\text{FL}]$–$O_2[\text{HW}]$. Each of these lines incurs 2 additional cross gates for preserving planarity through moving-out their gates outside of the $k$-times cascaded Figures 22 and 23. Consequently, our cellular automaton may conttain $5 \cdot 2 \cdot k(k-1)/2$ more cross gates than that in Lemma 2. The above analysis still holds even suffering this cross-gate incrementation. $\square$

## 5. Conclusions

We have shown that it is PSPACE hard to predict whether the ant residing in $h$-twist torus will visit almost all vertices or nearly none of them, for $h = \pm 11$. We have embedded a closed QCNF $\Phi$ to $\mathbb{Z}_n \times \mathbb{Z}_n$ of the twisted torus $\mathbb{Z}_N \times \mathbb{Z}_N$ and measured the following quantities by dividing $N^2$ and taking the limit $N \to \infty$. If $\Phi$ is TRUE, then the density of ever visiting vertices is 1, and the period is $\frac{208}{11}$. Otherwise, both are 0. The same argument proves the following result for any odd number $h \geq 11$. Let $h \bmod 11 := h - 11\lfloor h/11 \rfloor$. If $\Phi$ is TRUE, then the density is $1 - \frac{|h| \bmod 11}{|h|}$, and the period is $\frac{208}{11} \cdot (1 - \frac{|h| \bmod 11}{|h|})$ since we have Figure 24's screwing highway inserting $|h| \bmod 11$ spaces between $j$th and $(j+1)$th diagonals for every $j \in \lfloor j/11 \rfloor \cdot \mathbb{N}$. For example, for $|h| = 23$, Figure 24 paves two consecutive screw highways $H_1$ and $H_2$ by Figure 6. For $|h| \in \{1, 3, 5, 7, 9\}$, we can prove that the ant's reachability problem is PSPACE hard, since Figure 5's TURN makes [15]'s PSPACE completeness proof valid. However, we have not yet found any non-zero density trajectory of Langton's ant for $1 \leq |h| \leq 10$. For every even number $h$, particularly the standard torus $h = 0$, Langton's ant's known computational complexity is only the PTIME-hardness of Gajardo, Moreira, and Goles [14]. Proving PSPACE-hardness for even-twist torus is wide open.

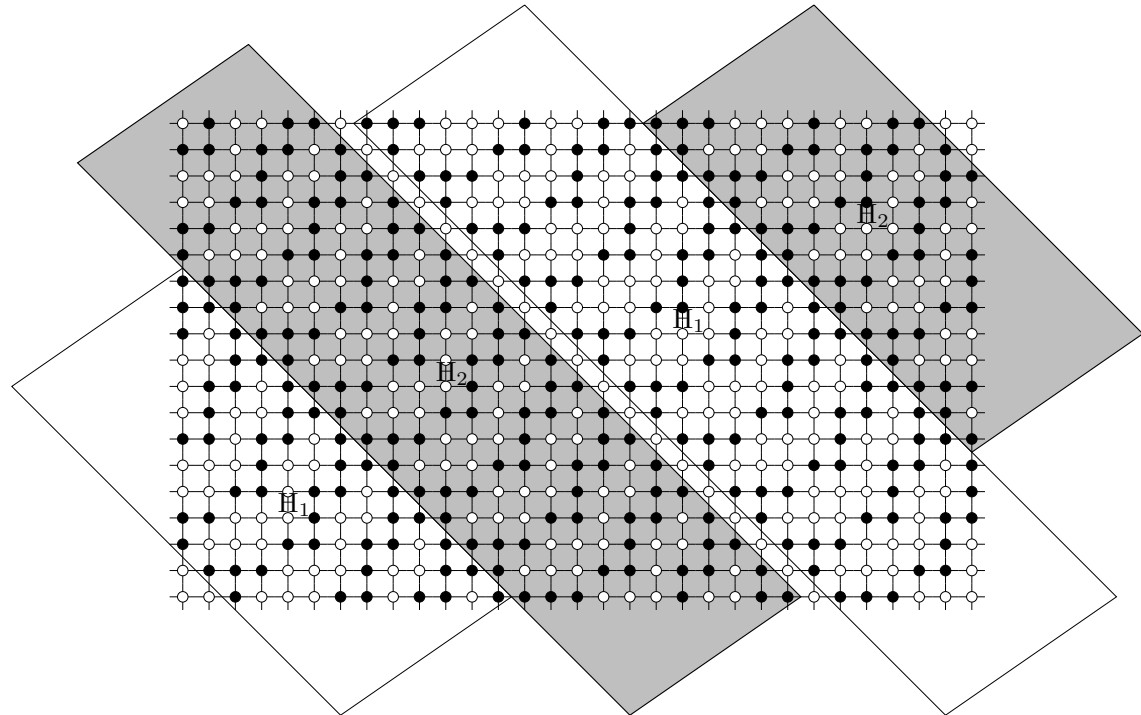

**Figure 24.** Screwing highways for $h = 23$.

**Author Contributions:** Conceptualization, T.T.; methodology, T.T.; software, T.H.; validation, T.H.; formal analysis, T.H.; investigation, T.H. and T.T.; writing—original draft preparation, T.T.; writing—review and editing, T.T.; visualization, T.H. All authors have read and agreed to the published version of the manuscript.

**Funding:** This research received no external funding.

**Conflicts of Interest:** The authors declare no conflict of interest.

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
