# Peer review of "Hardness of Approximation for Langton’s Ant on a Twisted Torus"

_algorithms, doi:10.3390/a13120344_

Round 1

Reviewer 1 Report

The aim of this study is to approximate the hardness of Langton’s ant on a twist torus. Generally this paper is really improved than last time that I studied. I have the following suggestions: 

1- Abstract should be improved. 

2- Again, I want to ask about the motivation of the authors for writing this paper. Add about motivation in Introduction and Abstract. 

3- Cite references for Theorem 1.  

4- The methodology is suitable. 

5- Change all figures to before Conclusion.

I propose a minor revision.   

Reviewer 2 Report

This is my second review of an article by T. Hagiwara and T. Tsukiji. The authors commented on the issues raised in the first one in detail. I consider all the answers sufficient. Thus, the article undoubtedly gained in quality. If anything could be improved in the current version, it would be the abstract, which from a very long one in the previous version of the article has become a concise one. Besides, the list of literature should be expanded, with particular emphasis on articles from recent years.

Author Response

This manuscript is a resubmission of an earlier submission. The following is a list of the peer review reports and author responses from that submission.

Round 1

Reviewer 1 Report

This article deals with some of the findings on Langton's ant on a twist torus.

The topic itself seems interesting, but its presentation is poor. The text is illegible, with numerous linguistic errors, which significantly hinders its understanding. The Reference list is very sparse and old, making it challenging to find additional information about the topic. Moreover, it is not entirely clear where are the authors' new results and where they are known from the literature.

Selected comments:
1) abstract: is too long (should be about 200 characters) and should not contain citations
2) the conclusion, in turn, can be expanded further
3) literature is cited in a chaotic order
4) figures are cited in a chaotic order
5) the quotation marks are not spelled correctly
6) line 26: there should be "about 10,000 iterations" (exactly 9977 according to [2])
7) n and N are used interchangeably (for example, in the term Z_n) - I have doubts as to whether such a designation is used correctly through the article everywhere
8) Fig. 3 should be rotated for legibility reasons
9) line 52: the definition of Phi is incomprehensible

Summing up, I do not recommend printing the article in its current form. I encourage the authors to rewrite the article, improve its style from the language, and end with the points I mentioned.

Reviewer 2 Report

The aim of this study is to approximate the hardness of Langton’s ant on a twist torus. I have the following suggestions to improve the quality of your study: 

1- The abstract is completely same with the introduction. I do not know why!! Please write the abstract again and describe about your aim, novelties and motivation. 

2- It would be better if you change the title to "Hardness Approximation of ...". 

3- There are so many serious English problems. 

4- Introduction is really week. I could not find anything about the application!! Also about novelties. There is no enough background in the introduction. 

5- In Line 3 change "An" to "an". Line 21, "The" to "the" and many others!!

6- I can not understand, why you need to find this approximation? What do you want to do after finding that??

7- After Line 40, describe about x,y, N. What are x,y, N?

8- Why you focused on h=+-11? 

9- What is the meaning of PSPACE in Line 43? You did not introduce before!! Also, QBF in Line 50. 

10- Lines 106, 141, change "Figure" to "Figures". 

11- Conclusion is really week and short. Rewrite again and describe the aim, novelties, motivation and also future plans!!

12- Update the list of your references. Cite new and recent papers.